First remains of the enormous alligatoroid Deinosuchus from the Upper Cretaceous Menefee Formation, New Mexico

Mohler Benjamin F. 1 benfmohler@email.arizona.edu
http://orcid.org/0000-0002-0324-0697 McDonald Andrew T. 2
http://orcid.org/0000-0003-2526-5059 Wolfe Douglas G. 3
1 Department of Geosciences, University of Arizona , Tucson, Arizona , United States of America
2 Western Science Center , Hemet, California , United States of America
3 Zuni Dinosaur Institute for Geosciences , Springerville, Arizona , United States of America
Wedel Mathew
Electronic publication date: 2021 Apr 21
Publication date: 2021
Volume: 9
Electronic Location ID: e11302
Received 2021 Jan 7; Accepted 2021 Mar 29
Copyright: © 2021 Mohler et al.
Copyright year: 2021
Copyright holder: Mohler et al.
License: This is an open access article distributed under the terms of the Creative Commons Attribution License, which permits unrestricted use, distribution, reproduction and adaptation in any medium and for any purpose provided that it is properly attributed. For attribution, the original author(s), title, publication source (PeerJ) and either DOI or URL of the article must be cited.
License URL: https://creativecommons.org/licenses/by/4.0/

Keywords: Deinosuchus, Menefee Formation, Campanian, Alligatoroid, Laramidia

Funding: Zuni Dinosaur Institute for Geosciences Southwest Paleontological Society The field work that led to discovery of the specimen was supported by small individual donations to the Zuni Dinosaur Institute for Geosciences and Southwest Paleontological Society. The funders had no role in study design, data collection and analysis, decision to publish, or preparation of the manuscript.

==============================
The neosuchian Deinosuchus is known from numerous localities throughout the Campanian of North America, from New Jersey to Montana (USA) and as far south as Coahuila (Mexico). Here we describe six osteoderms, two vertebrae, and a partial tooth discovered in the Menefee Formation of New Mexico and assign them to Deinosuchus sp., representing one of the earliest occurrences of this genus on the Laramidian subcontinent, and among the earliest known occurrences of this large alligatoroid in all of North America. The osteoderms are morphologically distinct in their inflated construction, with deep and radially distributed pitting, which closely matches osteoderms of Deinosuchus.

Introduction

The gigantic alligatoroid Deinosuchus is known from material on both sides of the Western Interior Seaway throughout the Campanian Stage in North America, giving the genus a wide temporal and geographic range. The holotype of the type species, Deinosuchus hatcheri, was reported from the Judith River Formation of Montana (Holland, 1909). Fossils belonging to the eastern species, Deinosuchus schwimmeri (Cossette & Brochu, 2020), have been reported from the Coffee Sand Formation of Mississippi and the Mooreville Formation of Alabama. Other eastern remains of Deinosuchus have been described from the Marshalltown Formation of New Jersey (Schwimmer, 2002), the Black Creek Formation of North Carolina (Schwimmer, 2002), the Blufftown Formation of Georgia and Alabama (Schwimmer, 2002), and the Demopolis Chalk of Alabama and Mississippi (Manning & Dockery, 1992; Schwimmer, 2002). A third species, Deinosuchus riograndensis (Colbert & Bird, 1954; Cossette & Brochu, 2020), is known from the Aguja Formation of Texas. Material referable to Deinosuchus sp. (Cossette & Brochu, 2020) has been found in the Fruitland Formation of New Mexico (Lucas, Sullivan & Spielmann, 2006; Sullivan & Lucas, 2006) and the Kaiparowits Formation of Utah (Titus et al., 2008; Irmis et al., 2013).

Beginning in 2011, yearly expeditions to exposures of the Upper Cretaceous Menefee Formation located in the San Juan Basin of northwestern New Mexico have been conducted by the Western Science Center, Zuni Dinosaur Institute for Geosciences, the Southwest Paleontological Society, and formerly the University of Pennsylvania (until 2013). This ongoing project, undertaken by professional researchers, students, and volunteers alike, has greatly enriched our understanding of this region of southern Laramidia during a poorly sampled interval, estimated to be around 80-79 Ma (middle Campanian). Recent publications include the descriptions of three new dinosaurs: the nodosaurid Invictarx zephyri (McDonald & Wolfe, 2018), the tyrannosaurid Dynamoterror dynastes (McDonald, Wolfe & Dooley, 2018), and the brachylophosaurin hadrosaur Ornatops incantatus (McDonald et al., 2021), while the descriptions of many other vertebrates, invertebrates, and plants recovered from these beds are underway.

Among these fossils are the remains of crocodyliforms, including WSC 285, the fragmentary remains of a large individual (or individuals) including several osteoderms that closely match those of Deinosuchus hatcheri (Holland, 1909). Lucas, Sullivan & Spielmann (2006) described Deinosuchus material from the Fruitland Formation of the San Juan Basin in northwestern New Mexico, stratigraphically higher than the Menefee, which was previously the only record of this taxon from the state. Though the remains of crocodyliforms have been known to occur in the Menefee Formation for decades (see the description of Brachychampsa sealeyi (Williamson, 1996)), it wasn’t until the recovery of WSC 285 from the Allison Member in 2018 that alligatoroids of substantial size were known to co-exist with dinosaurs among the coastal territories of mid-Campanian New Mexico.

Materials and Methods

WSC 285 was collected on land administered by the United States Bureau of Land Management (BLM) under permit NM 18-03S.

Geologic setting and fossil occurrence

The bones attributed to WSC 285 were collected from the Allison Member of the Menefee Formation (Sears, 1925; Miller, Carey & Thompson-Rizer, 1991; Molenaar et al., 2002) in the general vicinity where several additional vertebrate specimens have been recovered (descriptions in preparation). The Menefee Formation has traditionally been included in the Mesaverde Group (Pike, 1947; Beaumont, Dane & Sears, 1956) and much of the study of these strata has historically been directed toward potential coal resources (Hoffman, Campbell & Beaumont, 1993, and references within). Pike (1947) noted “one of the chief characteristics of this formation is the extreme irregularity of the individual beds”.

The transgressive marine Cliff House Sandstone intertongues with, and eventually supersedes, the terrestrial Menefee Formation; because of their complex stratigraphic relationship, any assessment of age and depositional environment of Menefee rocks near this boundary require that careful attention be paid to the Cliff House as well. Donselaar (1989) identified two principal units of the Cliff House Sandstone where the former coastline was roughly stationary, leading to thick vertically stacking deposits. The lowermost unit is sometimes called the La Ventana tongue and is best exposed in southern Colorado and in more eastern Cliff House Sandstone exposures near Cuba, New Mexico. Vertebrate fossils described from the Menefee Formation by Williamson (1996, 1997) were collected from below the La Ventana tongue and are stated to be lower Campanian. Donselaar (1989) provided a model for the Cliff House Sandstone near our locality as the product of stacked west-northwest to east-southeast directed elongate barrier bars, behind which the Menefee Formation mudstones and carbonaceous sediments were deposited.

The Campanian-age index fossil Baculites perplexus at localities within the Cliff House Sandstone near to, and along the depositional strike of, the study area (Siemers & King, 1974; Varela, Santucci & Tweet, 2019) provide a minimum age of 78.5 Ma for Menefee Formation vertebrate fossils. Furthermore, Lucas et al. (2005) obtained a radioisotopic date of 78.22 ± 0.26 Ma from high in the Menefee Formation in the Gallina hogback in Rio Arriba County, New Mexico, east of our study area in San Juan County.

The Allison Member (formerly called the “Allison barren member” (Sears, 1925; Beaumont, Dane & Sears, 1956)) lies above the Cleary Coal Member of the Menefee, and consists of three informal sub-units; the Lower Beds, the Juans Lake Beds, and La Vida Beds in ascending order (Miller, Carey & Thompson-Rizer, 1991). WSC 285 and nearby dinosaurs were recovered from the Juans Lake Beds. Alternatively, stratigraphic reconstructions of the Menefee Formation based on sites east of our study area (e.g., Williamson, 1997; Lucas et al., 2005; Dalman & Lucas, 2018; also see Fig. 2 in Beaumont & Hoffman (1992)) have divided the Menefee Formation into three members with an upper coal member composing the uppermost portion of the formation. However, the coal beds found in the upper portion of our study area do not have sufficient stratigraphic continuity to warrant designating a separate member (Dane, 1936; Miller, Carey & Thompson-Rizer, 1991).

The associated bones of WSC 285 were encountered as in situ elements and scattered float across several meters along a steeply gullied cliff face. A fragmentary neosuchian tooth was also collected as float from this layer. The two caudal vertebrae (WSC 285.7 and 285.8) and a large complete osteoderm (WSC 285.2) were recovered from an area of less than a square meter in blocky, olive gray to dark olive gray, silty mudstone. These were encased in a single plaster jacket and prepared at the WSC laboratory.

Lateral to the in situ bones lay large (>1 m) sideritic concretions, which, being much harder than surrounding mudstone layers, form locally prominent features influencing the steep and complex gully bisecting the fossil locality. Additional osteoderm fragments were collected from the base of the small gully and are presumed to have been weathered from the concretionary interval above. Fossil bone fragments and abundant plant debris were noted in large broken sections of the concretions. Several pyritized internal molds of gastropods and bivalves were collected as floated specimens immediately above the small “shelf” formed by the concretions.

The local stratigraphic sequence consists of highly variable, laterally discontinuous, laminar to blocky olive gray claystones, sandy-silty mudstones, grayish-black carbonaceous mudstones, lignites, and minor, generally discontinuous coal layers. A concretionary fine- to medium-grained sandstone supporting the top of the cliff above the locality appears to be one of the few stratigraphic units extending throughout the local outcrop belt, and locally divides the Deinosuchus locality below this sandstone from partial dinosaur skeletons (including the holotype of Dynamoterror dynastes (McDonald, Wolfe & Dooley, 2018), a new hadrosaurid (McDonald et al., 2021), and a partial ceratopsid skeleton) collected above this sandstone. Several additional vertebrate localities are currently under study within stratigraphic intervals several meters lower than the Deinosuchus locality.

Smaller sandstone stringers, minor sandstone channels, ribbon sands, cut-and-fill structures and soft-sediment deformation features are common throughout; these typically truncate more horizontal features such as coal layers which can rarely be traced laterally for more than a few tens or hundreds of meters. Large irregular sideritic concretions, some extending several meters as sinuous lenticular masses, are common, and these typically contain large quantities of carbonized and silicified plant debris, wood fragments, and rip-up clasts. Silicified wood, including large in situ stumps; carbonized logs and stumps; and widely distributed, transported, and abraded logs are abundant throughout the study area. This close association of terrestrial (dinosaurs, in situ trees), semi-aquatic (turtles, crocodilians) and aquatic organisms (bivalves, gastropods), combined with sedimentary features, suggests a forested floodplain interspersed with smaller rivers and coal swamps subject to periods of desiccation and flood.

The potential influence of coastal processes (tides, cyclones, earthquakes, tsunamis, etc.) upon more inland distributary systems cannot be discounted in a setting such as the Menefee Formation. Larger scale correlations indicate the subject locality lies within a few tens of miles westward of the western pinch-outs of correlative marine units (Molenaar et al., 2002; Varela, Santucci & Tweet, 2019). The potential influence of large-scale marine and coastal climate processes upon the terrestrial environments, paleoecology, and taphonomy of the Menefee Formation vertebrates is the subject of ongoing study.

Systematic Paleontology

Crocodylia Gmelin, 1789

Alligatoroidea Gray, 1844

Deinosuchus Holland, 1909

Deinosuchus sp.

Referred specimen: WSC 285, incomplete associated postcranial remains including six osteoderms and two caudal vertebrae, as well as one fragmentary tooth. Digital 3-D models of WSC 285 are available on MorphoSource (Project name: “Menefee Deinosuchus”) and Sketchfab.

Location: Collected in San Juan County, New Mexico, on land administered by the United States Bureau of Land Management (BLM). Precise locality data are on file at WSC and the BLM.

Horizon: Collected in the Juans Lake Beds, Allison Member, Menefee Formation (Miller, Carey & Thompson-Rizer, 1991); middle Campanian, Upper Cretaceous (Molenaar et al., 2002; Lucas et al., 2005).

Description. Deinosuchus is distinct among Campanian crocodilians in exhibiting large osteoderms with notably thick cross-sections, giving them an “inflated” appearance. This inflation has a considerable range, as noted by Holland (1909), who characterized medium-sized osteoderms as “almost hemispherical” and the smallest osteoderms as “spherical”. Smaller to medium-sized osteoderms can be weakly keeled, a trait that disappears with additional bony overgrowth in larger specimens (Schwimmer, 2002). Pitting in these osteoderms is circular, deep, and radially distributed; the depth and irregularity of pitting varies within the species of Deinosuchus, with increasing depth and increasing irregularity associated with increasing osteoderm size (Cossette & Brochu, 2020). As established by Holland (1909) and reaffirmed by subsequent descriptions of Deinosuchus fossils (Colbert & Bird, 1954; Schwimmer, 2002; Lucas, Sullivan & Spielmann, 2006; Cossette & Brochu, 2020), alligatoroid osteoderms can be assigned to Deinosuchus based on these features. No described specimen of Deinosuchus has preserved the entirety of the nuchal, dorsal, and sacrocaudal shields, making comparison of osteoderm morphology between species difficult (see “Discussion”).

WSC 285 comprises an incomplete postcranium recovered from a single locality within the Juans Lake Beds of the Allison Member of the Menefee Formation. Material includes two complete osteoderms, one nearly complete osteoderm, and three additional osteoderm fragments (Fig. 1). WSC 285.1, the smaller of the two complete osteoderms (Figs. 1A–1C), likely represents a nuchal or sacrocaudal osteoderm due to its smaller size, rounded edges, and cross-sectional thickness approaching the total width (Schwimmer, 2002; also see Fig. 4H (CM 963) in Cossette & Brochu (2020), but also note on WSC 285.1 the lack of marginal indentation that possibly distinguishes D. hatcheri from other species). It is weakly keeled and highly inflated, displaying deep tear-shaped pitting- similar to the smaller, oval-shaped, and keeled osteoderms of D. schwimmeri that Cossette & Brochu (2020) assigned to the nuchal shield. The larger complete osteoderm, WSC 285.2 (Figs. 1D–1F), measures 7.5 cm long and 11.1 cm wide at the most extreme margins, and 3.6 cm thick from its ventral surface to the apex of its keel. It shows similarities to osteoderms of D. riograndensis that Cossette & Brochu (2020) assigned a caudal position along the dorsal shield, due to width greatly exceeding length and a keel that skews slightly towards the lateral margin (see Figure 17I-J (TMM43632-1) in Cossette & Brochu (2020)). It also features a distinct cross-hatching texture on the ventral surface (Fig. 1E), a feature which Cossette & Brochu (2020) considered strongly indicative of dorsal placement. The large partial osteoderm, WSC 285.3 (Figs. 1G, 1H), also likely represents a component of the dorsal shield, showing similarities to the large and rectangular osteoderms of D. schwimmeri that Cossette & Brochu (2020) assigned to the dorsal shield (see the leftmost osteoderm in Figure 25J (ALMNH 1002) in Cossette & Brochu (2020)). Pitting on these larger osteoderms is more circular and similarly deep. Additional osteoderm fragments were recovered which exhibit the same inflated and deeply pitted structure (Figs. 1I–1L). Definitive Deinosuchus osteoderms, such as with the holotype of Deinosuchus hatcheri (CM 963 (Fig. 2), and also see Fig. 4 in Cossette & Brochu (2020)), show a considerable range in density of pitting as well as overall shape, even among specimens with finished edges; however, specimens show the same combination of a typically thick cross-section (though eastern Deinosuchus schwimmeri can be much thinner), deep pitting, and inflation of the keel, if a keel is present.

Figure 1 Osteoderms of WSC 285.

Osteoderms of Deinosuchus sp., including nuchal or sacrocaudal osteoderm WSC 285.1 in dorsal (A), ventral (B), and cranial or caudal (C) views; dorsal osteoderm WSC 285.2 in dorsal (D), ventral (E), and cranial or caudal (F) views; dorsal osteoderm WSC 285.3 in dorsal (G) and ventral (H) views; fragmentary osteoderms WSC 285.4 and WSC 285.5 in dorsal view (I and J); and fragmentary osteoderm WSC 285.6 in dorsal (K) and ventral (L) views. All scale bars = 5 cm.

Figure 2 Holotype osteoderms of Deinosuchus hatcheri.

CM 963, various osteoderms from the holotype of Deinosuchus hatcheri in dorsal (A–C) views. Scale bar = 5 cm.

The site also yielded two nearly complete caudal vertebrae. The more complete of the two (WSC 285.7; Fig. 3) measures 11.2 cm along the craniocaudal length of the centrum, and 15.0 cm from the ventral surface of the centrum to the top of the neural spine. The neural spine is largely complete, with an elongated base extending nearly the full length of the centrum and tapering to around half its basal length at its broken apex. The left transverse process is nearly complete, extending 8.1 cm from its origin at the centrum to its tip. This process projects ventrolaterally with a moderate caudal inclination. The left postzygapophysis is well-preserved, originating in the caudal region of the base of the neural spine and projecting caudolaterally. The other caudal vertebra is less complete but approximately the same size (WSC 285.8; Fig. 4), measuring 11.3 cm from end to end. It is strongly procoelous and both articular surfaces are well-preserved. The ventral surface of the centrum is gently concave in lateral view. The ventral surface of the centrum exhibits two parallel ridges extending craniocaudally, with a depression in between. The transverse processes are broken near their bases. The neural arch is intact but the neural spine is not preserved beyond the first few centimeters that make up its base.

Figure 3 Caudal Vertebra of WSC 285.

Caudal vertebra of Deinosuchus sp., WSC 285.7, in cranial (A), left lateral (B), and caudal (C) views. Abbreviations: ns, neural spine; poz, postzygapophysis; tp, transverse process. All scale bars = 5cm.

Figure 4 Caudal Vertebra of WSC 285.

Caudal vertebra of Deinosuchus sp., WSC 285.8, in left lateral (A), right lateral (B), dorsal (C), ventral (D), cranial (E), and caudal (F) views. Abbreviations: tp, transverse process. All scale bars = 5 cm.

A single shed neosuchian tooth was also recovered in association with the Deinosuchus postcrania (Fig. 5). Though highly fragmented, portions of the crown and root are present. While we do not assign WSC 285 beyond the generic level (see Discussion), the thickness of tooth enamel has been proposed as potentially diagnostic to Deinosuchus and as a potential differentiator between eastern and western ‘morphs’ of Deinosuchus (see Schwimmer, 2002). However, the lack of definitive parameters for this kind of evaluation, along with newly introduced osteological differentiators between D. riograndensis and D. schwimmeri (Cossette & Brochu, 2020) means that this method has fallen out of favor.

Figure 5 Associated crocodyliform tooth.

Tooth found associated with WSC 285 in cross-sectional (A) and external (B) views. Scale = 5 mm.

Discussion

Though osteoderm morphology appears to be sufficient for assignment of neosuchian material to Deinosuchus, it is important to consider cases where assignment was unclear or has since been disputed. Foster & Hunt-Foster (2015) reported a neosuchian osteoderm from the Williams Fork Formation of Colorado, at around 74.6–72.7 Ma (Fowler, 2017). This specimen, MWC 8240, is a large osteoderm resembling those of Deinosuchus in its deep pitting, but atypical in that it is not inflated and appears to be more subrectangular. Osteoderms morphologically similar to MWC 8240 but smaller in size are abundant in the Williams Fork Formation (Foster & Hunt-Foster, 2015), indicating an abundance of this morph as juveniles, or perhaps that few adults achieved maximum size. Whether this represents a new species of Deinosuchus or a separate genus of neosuchian is unclear, and therefore a diagnosis beyond Neosuchia was not made (Foster & Hunt-Foster, 2015). Similar to MWC 8240 is a specimen reported from the Mesaverde Formation of Wyoming, assigned directly to Deinosuchus sp. (Wahl & Hogbin, 2003). This specimen, UW 16040, is an osteoderm of similar construction to MWC 8240 in that it displays deep pitting and large size typical of Deinosuchus, but differs from Deinosuchus in its subrectangular shape and thin cross-section. Wahl & Hogbin (2003) argued that the thin cross-section is a taphonomic artefact resulting from erosion of the ventral surface, though Foster & Hunt-Foster (2015) considered this unlikely.

Cossette & Brochu (2020) revised the taxonomy of Deinosuchus and recognized three species: D. hatcheri, D. riograndensis, and D. schwimmeri. The latter two species, D. riograndensis and D. schwimmeri, are known from more complete material and are distinguished primarily using characters present in cranial material, which WSC 285 lacks for comparison. Osteoderm morphology is also a suggested differentiator, with D. hatcheri displaying the highest degree of keel inflation along with lumpy and overall irregular shape; D. riograndensis displaying less exaggerated keel inflation but invariably lumpy and overall irregular shape; and D. schwimmeri displaying exaggerated keel inflation (but variable degree of inflation and shape of the keel) and more regular overall shape (Cossette & Brochu, 2020). A feature considered possibly diagnostic for D. hatcheri by Cossette & Brochu (2020) is a unique indentation along one edge of dorsal osteoderms, which might indicate placement along the lateral edge of the dorsal shield. None of the six complete or partial osteoderms of WSC 285 exhibit such an indentation, thereby precluding referral to D. hatcheri. Nevertheless, the osteoderms of WSC 285 are consistent with specimens of the three species of Deinosuchus in having a rounded to subglobose shape, inflated keel, and deep circular pitting, which we consider sufficient for assignment to Deinosuchus; a taxonomic assignment no more specific than Deinosuchus sp. is most prudent at this time.

Although WSC 285 probably ranks among the oldest examples known for the genus Deinosuchus, resolving an exact age for any material recovered from the Menefee Formation is difficult. Based upon correlation with marine biostratigraphic zones, Molenaar et al. (2002) showed a range of approximately 84.0-78.5 Ma for the formation. The occurrence of the ammonite index fossil Baculites perplexus in the overlying Cliff House Sandstone near our field area indicates that WSC 285 must be older than 78.5 Ma (Siemers & King, 1974; Molenaar et al., 2002). Lucas et al. (2005) determined a more precise age of 78.22 ± 0.26 Ma for the upper reaches of the Menefee Formation based on radioisotopic data from the Gallina hogback, located in the eastern San Juan Basin. However, this date is in conflict with palynostratigraphic data reported in the same study: pollen recovered from carbonaceous mudstone throughout Menefee exposures at the Gallina hogback yielded an assemblage similar to late Santonian to earliest Campanian pollen from the Milk River, Telegraph Creek, and lower Eagle Formations farther north in Laramidia (Lucas et al., 2005). The possibility of a transgressive unconformity at the base of the Cliff House Sandstone, as well as potential issues with extrapolating age based on palynomorphs from as far north as Alberta, Canada, means that more intensive study is needed before this discrepancy can be explained (Lucas et al., 2005).

Within the Allison Member are three informally recognized subdivisions: from lowest to highest, these are the Lower Beds, the Juans Lake Beds, and the La Vida Beds (Miller, Carey & Thompson-Rizer, 1991). Deinosuchus sp. (WSC 285) is herein reported from the Juans Lake Beds (Miller, Carey & Thompson-Rizer, 1991), high in the Allison Member of the Menefee Formation but well below the Cliff House Sandstone in which Baculites perplexus occurs (Siemers & King, 1974), leading to a tentative age estimation of 80-79 Ma. Elsewhere in the San Juan Basin of northwestern New Mexico, Deinosuchus sp. has previously been recovered from the Fossil Forest Member of the Fruitland Formation (Lucas, Sullivan & Spielmann, 2006; Sullivan & Lucas, 2006; Cossette & Brochu, 2020) which is dated to a narrow window of 75.4 to 75.2 Ma (Fowler, 2017).

Material comparable in age to WSC 285 has been found in the lower to middle Campanian Aguja Formation of Texas. The lower shale member of the Aguja Formation, which is constrained to between 82 and 80.5 Ma (Fowler, 2017), has produced an osteoderm tentatively assigned to cf. Deinosuchus sp. by Lehman et al. (2019). The younger upper shale member of the Aguja Formation, between 80 and 77 million years old, contains Deinosuchus in higher abundance (Lehman et al. (2019); assigned to D. riograndensis by Cossette & Brochu (2020)), while the upper shale member of the Aguja Formation in neighboring Coahuila, northern Mexico, has produced a partial left surangular, teeth, and postcranial material including osteoderms (Rivera-Sylva & Frey, 2011; Rivera-Sylva et al., 2011). The lower-to-middle Campanian Wahweap Formation of Utah, roughly similar in age to the Allison Member, thus far lacks any material of Deinosuchus (Irmis et al., 2013), while the overlying Kaiparowits Formation, at about 76.5 to 74.5 Ma (Fowler, 2017), has produced material referable to Deinosuchus sp. (Titus et al., 2008; Irmis et al., 2013; Cossette & Brochu, 2020). The holotype of Deinosuchus, CM 963, was found farther north in the Judith River Formation of Montana, at approximately 79.5 to 75 Ma (Holland, 1909; Lucas, Sullivan & Spielmann, 2006; Fowler, 2017).

To the east, Deinosuchus schwimmeri has been reported from the Mooreville Formation of Alabama (Cossette & Brochu, 2020), which is between 84 and 79.5 Ma (see Fig. 1 in Puckett, 2005). This represents the oldest currently known remains of Deinosuchus, estimated at 82 Ma (Schwimmer, 2002). The Blufftown Formation in Alabama and Georgia, which correlates to the Mooreville Formation in Alabama (Sohl, 1964), has also produced Deinosuchus fossils dating to between 82 and 79 Ma (Schwimmer et al., 1993; Schwimmer, Stewart & Williams, 1994; Fig. 4.6 in Schwimmer (2002)), itself closely corresponding in age to the Allison Member of the Menefee Formation. Schwimmer (2002) reported Deinosuchus from the Demopolis Chalk in Alabama, between 79.5 and 74 Ma (see Fig. 2 in Prieto-Márquez, Erickson & Ebersole (2016)); Manning & Dockery (1992) also reported Deinosuchus from the Demopolis Chalk in neighboring Mississippi. Deinosuchus schwimmeri was recently named based upon material from the Coffee Sand Formation of Mississippi (Cossette & Brochu, 2020), a formation which correlates to the Blufftown Formation of Alabama and Georgia in its lower portion and the Demopolis Chalk of Mississippi and Alabama in its upper portion (Sohl, 1964). Cossette & Brochu (2020) referred to the Coffee Sand Formation as middle Campanian, constraining it to between approximately 81 and 76.5 Ma according to Fowler (2017). Fossils of Deinosuchus have also been recovered from the Black Creek Formation of North Carolina, which is between 80 and 71 Ma (see Fig. 4.6 in Schwimmer (2002)); however, Schwimmer (2002) reported that eastern Deinosuchus appear to go extinct in the southeastern United States by about 77 Ma. Finally, Deinosuchus is also reported from the Marshalltown Formation of New Jersey (Schwimmer, 2002), which dates to between 79.6 and 76.4 Ma (Denton & Tashjian, 2012; Brownstein, 2018).

Therefore, WSC 285 from the Allison Member of the Menefee Formation represents one of the oldest records of Deinosuchus from Laramidia, and is likely within a few million years of the oldest occurrences in Appalachia. This indicates an early and wide distribution of the genus throughout the southern regions of both subcontinents (New Mexico, Texas, Coahuila, Alabama, Georgia, and Mississippi) by the middle Campanian (Fig. 6). More intensive sampling of Campanian rocks in these regions, along with more precise dating, may make it possible to discern which region the genus originally evolved in and what route was taken to spread across North America. Though Deinosuchus was certainly capable of travelling through aquatic environments as well as on land, it is unlikely that this radiation was achieved directly by swimming across the Western Interior Seaway (Schwimmer, 2002). Wells drilled in the southern portion of the former Seaway have recovered pyroclastics, lava flows, and breccia-filled plugs that point to the presence of a volcanic island chain in the region, which might have provided a terrestrial or near-shore route of dispersion for Deinosuchus, as well as for other fauna (Carpenter, 1982). Alternatively, Cossette & Brochu (2020) posited that radiation of an ancestral alligatoroid before the transgression of the Western Interior Seaway is likely, leading to allopatric speciation among early populations of Deinosuchus.

Figure 6 Biogeography of Deinosuchus.

Dashed bars indicate either uncertainty in the ages of Deinosuchus-bearing units or portions of units in which Deinosuchus has not been documented. The skeletal reconstruction of Deinosuchus used here was provided by the artist, Tyler Holmes, and is used with permission. The map of North America was created by approximately tracing the paleogeographic map of the Western Interior Seaway created by the Cretaceous Atlas of Ancient Life (https://www.cretaceousatlas.org/geology/), licensed under a Creative Commons Attribution-NonCommercial-ShareAlike 4.0 International License (https://creativecommons.org/licenses/by-nc-sa/4.0/deed.en_US). Data for the Coffee Sand Formation occurrence comes from Cossette & Brochu (2020) and Fowler (2017); for the Demopolis Chalk, from Manning & Dockery (1992), Schwimmer (2002), and Prieto-Márquez, Erickson & Ebersole (2016); for the Mooreville Formation, from Schwimmer (2002), Puckett (2005), and Cossette & Brochu (2020); for the Blufftown Formation, from Sohl (1964), Schwimmer et al. (1993), Schwimmer, Stewart & Williams (1994), and Schwimmer (2002); for the Black Creek Formation, from Schwimmer (2002); for the Marshalltown Formation, from Schwimmer (2002), Denton & Tashjian (2012), and Brownstein (2018); for the Judith River Formation, from Holland (1909), Lucas, Sullivan & Spielmann (2006), and Fowler (2017); for the Aguja Formation, from Rivera-Sylva & Frey (2011), Rivera-Sylva et al. (2011), Fowler (2017), and Lehman et al. (2019); for the Menefee Formation, from Dane (1936), Siemers & King (1974), Miller, Carey & Thompson-Rizer (1991), Beaumont & Hoffman (1992), Molenaar et al. (2002), and Lucas et al. (2005); for the Fruitland Formation, from Sullivan & Lucas (2006), Lucas, Sullivan & Spielmann (2006), Fowler (2017), and Cossette & Brochu (2020); and for the Kaiparowits Formation from Titus et al. (2008), Irmis et al. (2013), Fowler (2017), and Cossette & Brochu (2020). Abbreviations are as follows: MS, Mississippi; AL, Alabama; GA, Georgia; NC, North Carolina; NJ, New Jersey; MT, Montana; TX, Texas; CH, Coahuila; NM, New Mexico; UT, Utah.

Eastern remains of Deinosuchus have been recovered from marine beds, suggesting an association with saltwater. However, due to the nearly complete lack of terrestrial beds representing the Appalachian landmass during the Campanian, it is unclear whether these animals lived in near-shore marine environments or were merely deposited in them (Schwimmer, 2002). In contrast, Laramidian Deinosuchus have been found in terrestrial beds, such as the Fruitland, Kaiparowits, Judith River, and Aguja formations (Holland, 1909; Colbert & Bird, 1954; Lucas, Sullivan & Spielmann, 2006; Irmis et al., 2013; Lehman et al., 2019; Cossette & Brochu, 2020). Rivera-Sylva et al. (2011) specifically reported Deinosuchus material from deltaic brackish facies of the Aguja Formation in Coahuila, Mexico. Very few fossils of D. riograndensis and D. schwimmeri have been found in deep-water deposits despite favorable preservation conditions existing there, further suggesting that the large alligatoroid did not often enter open water (Cossette & Brochu, 2020). Furthermore, Deinosuchus has no clear adaptations for processing and secreting excess salt (Cossette & Brochu, 2020).

WSC 285 was recovered from a site geographically and stratigraphically among numerous dinosaur and freshwater trionychid turtle localities. This, along with the aforementioned freshwater bivalves present at the field site, indicate that WSC 285 is another example of Deinosuchus from a terrestrial freshwater setting. However, detailed stratigraphic and sedimentological data for this and other localities in our field area are required to fully elucidate the depositional history of the Allison Member; Lewis et al. (2007) reported a vertebrate microsite in the Allison Member in the eastern San Juan Basin which preserves a mixed fauna of terrestrial and marine taxa, stressing the need for close examination of these beds before drawing strong paleobiological conclusions. This work is ongoing and will be published elsewhere.

It is also worth considering available food sources when evaluating the distribution of Deinosuchus. It has been suggested by Cossette & Brochu (2020) that the larger body size of the western species Deinosuchus riograndensis over the eastern species Deinosuchus schwimmeri may have been due to more favorable climate, more abundant prey, or both. Further paleoecological and paleoclimatological reconstructions of conditions during deposition of the Allison Member may help contribute to a better understanding of coastal Western Interior productivity and climate, which in turn would help test that hypothesis. To date, dinosaur faunas from the few “middle” Cretaceous terrestrial strata deposited during maximum sea-level rise are composed of smaller members of well-known groups that reached larger sizes by the Campanian (Nesbitt et al., 2019). As seas retreated, coastal areas expanded, and as forests became larger so did the herbivores and predators dependent upon them. The middle Campanian Menefee Formation dinosaurs are much larger than the Turonian dinosaurs of the Moreno Hill Formation in west-central New Mexico, within comparable clades (ceratopsians, tyrannosauroids, hadrosauroids). The size and distribution of Deinosuchus may thus reflect the increased size and abundance of prey within expanding coastal environments.

Conclusions

Fragmentary associated remains of Deinosuchus sp. are reported from the Allison Member of the Menefee Formation, which extends the temporal record of this alligatoroid in the San Juan Basin of northwestern New Mexico back to the middle Campanian. While it was previously known that Deinosuchus was widely distributed in southern Appalachia (now the southeastern United States) around this time, and tentatively suggested that it had reached as far westward as Texas, it is now known definitively that the genus was also present within the coastal ecosystems of southern Laramidia by the middle Campanian.

Photographs of the Deinosuchus holotype (CM 963) were provided by Amy Henrici (Carnegie Museum of Natural History) and are used with permission. Photographs used in Fig. 5 were provided by Mark Fredricksen and Brett Dooley (Western Science Center). The authors would also like to recognize the contributions of co-discoverer Sherman Q. Mohler, who through his years of leadership at the Southwest Paleontological Society has provided critical financial and logistical support to the Menefee Project for a full decade. Fossils were discovered by Mark Howard and Sherman Q. Mohler, collected by Mark Howard, Kara A. Kelley, Andrew T. McDonald, Benjamin F. Mohler, Sherman Q. Mohler, Joe Reavis, and Douglas G. Wolfe. Preparation at the Western Science Center was carried out by Leya Collins, Darla Radford, and Joe Reavis. The digital 3-D models were created at Western Science Center by Alton Dooley and Brett Dooley. We thank the reviewers, Joshua Cohen, Joseph Frederickson, and David Schwimmer, for their comments that improved the manuscript.

Institutional Abbreviations

ALMNH Alabama Museum of Natural History, Tuscaloosa, Alabama, USA

CM Carnegie Museum of Natural History, Pittsburgh, Pennsylvania, USA

MWC Museum of Western Colorado, Fruita, Colorado, USA

TMM Texas Memorial Museum, Austin, Texas, USA

UW University of Wyoming Collection of Fossil Vertebrates, Laramie, Wyoming, USA

WSC Western Science Center, Hemet, California, USA

Additional Information and Declarations

Competing Interests

Author Contributions

Field Study Permissions

Data Availability

The authors declare that they have no competing interests. Benjamin F. Mohler is employed by the University of Arizona, Tucson, Arizona; Andrew T. McDonald is employed by Western Science Center, Hemet, California; and Douglas G. Wolfe is employed by Zuni Dinosaur Institute for Geosciences, Springerville, Arizona.

Benjamin F. Mohler performed the experiments, analyzed the data, prepared figures and/or tables, authored or reviewed drafts of the paper, and approved the final draft.

Andrew T. McDonald conceived and designed the experiments, performed the experiments, analyzed the data, authored or reviewed drafts of the paper, and approved the final draft.

Douglas G. Wolfe conceived and designed the experiments, performed the experiments, analyzed the data, authored or reviewed drafts of the paper, and approved the final draft.

The following information was supplied relating to field study approvals (i.e., approving body and any reference numbers):

Fieldwork was conducted under permit issued by the U.S. Bureau of Land Management (Permit NM 18-03S).

The following information was supplied regarding data availability:

Fossils are available at the Western Science Center (WSC), ID WSC 285.1-285.8

The 3D files are available at MorphoSource:

DOI 10.17602/M2/M348667.

DOI 10.17602/M2/M348663.

DOI 10.17602/M2/M348659.

DOI 10.17602/M2/M348655.

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
