# Peer review of "First remains of the enormous alligatoroid Deinosuchus from the Upper Cretaceous Menefee Formation, New Mexico"

_PeerJ, doi:10.7717/peerj.11302_

## Round 0.1 · original submission · Minor Revisions

The reviewers found much of value in your manuscript, and all provided constructive suggestions. Please note that, contra reviewer 1, perceived impact or importance are not editorial criteria at PeerJ, so long as the manuscript is scientifically sound. With that said, I believe reviewer 1 has some valuable suggestions for improving the impact of your work. The reviewers' other comments all seem apt. In particular, please note the reviewers' comments regarding the oldest Deinosuchus material from both Laramidia and Appalachia, and also their concerns about whether you've documented sufficient comparative data to support your referral.

I look forward to seeing an improved version of this work in the near future.

·

Basic reporting

The geologic setting and fossil occurrence section requires a reorganization to better flow logically. The Systematic Paleontology section should not be split up. See specific comments on attached pdf.

Experimental design

The research question is not well defined or meaningful. A new specimen of Deinosuchus is identified without much discussion on its importance. The authors attempt to argue that this specimen is one of the oldest specimens in Laramidia, but do not comment on why this is important on the evolution of this taxon. See specific comments on attached pdf.

Validity of the findings

Conclusions are a bit confusing and not well supported by the data. The authors should expand their discussion to include more robust comparisons to scutes to all species of Deinosuchus and to other closely related taxa. See specific comments on attached pdf.

Additional comments

See specific comments on attached pdf.

·

Basic reporting

The article is fairly simple and straightforward: they report a new occurrence of a well-known organism represented by a few osteoderms, two vertebrae and a fragmentary tooth. The major new detail in the article is the claim that this is an older occurrence on the western side of the continent. I have no disagreement with the technical details, except regarding some of the terminology for descriptions and age-ranges, detailed in the "validity" discussion below.
The formatting, image quality, etc. seem fine. I would have preferred an additional image in Fig. 2 to show the thickness of the holotype osteoderm for comparison.

Experimental design

This is an observational report so "design" is not really relevant. It follows traditional formatting. The overall material is acceptable with a few minor issues below.

Validity of the findings

There is no doubt that the specimen reported are Deinosuchus osteoderms and vertebrae. However, I am not convinced they are very different from typical Deinosuchus riograndensis osteoderms (the vertebrae and tooth are nondescript). The authors here seem to imply they conform more closely with those of D. hatcheri based on "inflation." However that is not what I observe in their figure 1Cand 1F, and it is notable they do not figure a side view of the holotype D. hatcheri material to provide a comparison. I would like a bit more detailed measurement and imaging of this.
Additionally, the age issue is interesting. Accepting their age for this occurrence at 79 Ma, that may indeed be old for the Laramida subcontinent, but it is 3 Ma younger than the oldest reported eastern occurrence. Three Ma is a significant difference which they gloss over by reporting their material as "roughly 80 million years (line 362)," and indicating age equivalence in their fig. 6. I believe this is significant when we try to figure out the origins of Deinosuchus species: if they are all equally old occurrences it suggest a different pattern than if one population on the various sides of the seaway is older than the other. By selecting 79 Ma as the boundary of "older" and "middle" age populations, they may create a false impression of simultaneous appearances. I would like it to be explicitly pointed out that this old Laramida occurrence is still notably younger than the oldest in Appalachia.

Additional comments

A few editing suggestions:

line 53 and elsewhere: there is no "middle Campanian" so I suggest "mid-Campanian" as the proper usage.
line 148 and elsewhere "in-situ" is thus.
line 171: the citations are in no order. Should be by date sequence.
line 189: cross-hatching misspelled
line 199: I'm not sure "inflated keel" is the correct term. Many 'derms lack keels at all, especially in the D. hatcheri type. Also, many osteoderms from Appalachia are quite thin (see Cossette and Brochu 2020, fig 26)
line 292: please note my comments about the age of this specimen. Note that 79 Ma does not "closely overlap" with the oldest Appalachian occurrence. This creates a false equivalency.
line 360: the term "associated skeleton" is a bit exaggerated for 6 'derms, 2 vertebrae and a fragment of tooth. This association is about 2% of the skeleton.
References: I notice there are no pages cited for Schwimmer (2002). it's 219 p.

·

Basic reporting

The manuscript was well-written and generally a good review for the limited scope of the paper. I see no issues with the structure of the manuscript or references included therein. There are a few errors or stylistic points outlined below:

Line 20: add "partial" to “tooth".
Line 65: delete a ")".
Line 117: check style whether in situ should be italicized.
Line 122 (throughout) check spacing between measurements and units.
Line 201: "from one end of the centrum to the other” craniocaudal?

Experimental design

The experiment in this case is a comparison between scutes. Though I believe the authors have a strong argument for their identification, the comparisons are fairly short and qualitative. In fairness, this is not the author’s fault and directly related to the scrappy nature of the Deinosuchus fossil record. Also on line 215, the authors briefly introduce the tooth and mention that previous authors have used enamel thickness to identify species (though this criteria was recently rejected by Cossette and Brochu, 2020). The authors state “...an evaluation of enamel can be taken.”, but don’t provide measurements or comparisons on this matter.

Validity of the findings

I have no issues with the findings and conclusions. I believe the authors are cautious not to overinterpret their results.

Additional comments

Overall, it is a good manuscript that provides another data point for one of the most interesting Cretaceous species from North America.

---

## Round 0.2 · accepted · Accept

Thank you for your diligence in addressing the concerns of the reviewers. I am satisfied with the revised manuscript, and I am happy to accept it for publication in PeerJ.

It was kind of you to thank the reviewers in the Acknowledgements. It was also kind of you to thank me as editor, but quite unnecessary. I've been happy to handle this paper, but nothing I've done has gone beyond the normal scope of the job, and if I get a say, I'd prefer not to be acknowledged. It's nothing against your paper -- as handling editor, I'll be linked to it for all time, and I wouldn't send an 'Accept' decision if I wasn't happy with that arrangement. Rather, I think formal acknowledgment should be for substantial contributions only, beyond what little I have done.

The decision of whether or not to publish the peer reviews alongside the paper is entirely yours, and will not affect how your paper is handled going forward. However, I encourage you to do so. Making the reviews public allows the reviewers to receive credit for their efforts, and also contributes to the emerging culture of fairness and transparency in editing and peer review.